# Genetic Polymorphisms (ApaI, FokI, BsmI, and TaqI) of the Vitamin D Receptor (VDR) Influence the Natural History and Phenotype of Crohn’s Disease

**DOI:** 10.3390/ijms26051848

**Published:** 2025-02-21

**Authors:** Theodora Kafentzi, Efthymios P. Tsounis, Evanthia Tourkochristou, Evdoxia Avramopoulou, Ioanna Aggeletopoulou, Georgios Geramoutsos, Christos Sotiropoulos, Ploutarchos Pastras, Konstantinos Thomopoulos, Georgios Theocharis, Christos Triantos

**Affiliations:** Division of Gastroenterology, Department of Internal Medicine, Medical School, University of Patras, 26504 Patras, Greece; dora_kafentzi@hotmail.com (T.K.); makotsouno@gmail.com (E.P.T.); evanthiatourkohristou@gmail.com (E.T.); eudo_avra@windowslive.com (E.A.); iaggel@upatras.gr (I.A.); giorgosgeramoutsos@gmail.com (G.G.); cr.sotiropoulos@hotmail.com (C.S.); ploutarchosp96@gmail.com (P.P.); kxthomo@hotmail.com (K.T.); georgiostheocharis@hotmail.com (G.T.)

**Keywords:** genetic polymorphisms, SNPs, inflammatory bowel disease, Crohn’s disease, vitamin D receptor, VDR

## Abstract

Vitamin D receptor (VDR) single-nucleotide polymorphisms (SNPs) modulate vitamin D/VDR signaling, a key pathway in inflammatory bowel disease (IBD) pathogenesis. This study investigates how ApaI, BsmI, TaqI, and FokI SNPs affect IBD phenotype and progression. A total of 76 Crohn’s disease (CD) and 68 ulcerative colitis (UC) patients were genotyped. On initial bivariate analysis, the AA genotype of ApaI was accompanied by higher rates of penetrating (B3) CD (36.7% vs. 8.7%; *p* = 0.012). The FokI SNP was associated with disease location, with the ff genotype predisposing to CD and affecting the upper GI (36.4% vs. 7.7%; *p* = 0.044) or the colon (90.9% vs. 50.8%; *p* = 0.038). Moreover, patients harboring the ApaI A allele (AA/Aa) experienced higher rates of steroid-refractory or steroid-dependent CD. In multivariate analyses, the aa genotype showed a protective effect against hospitalization (aOR = 0.17; *p* = 0.013) in CD, whereas the TT genotype emerged as an independent risk factor (aOR = 4.79; *p* = 0.044). Moreover, the aa genotype was independently associated with a decreased risk of IBD-related surgery (aOR = 0.055; *p* = 0.014). VDR SNPs, particularly ApaI, influence disease phenotype, progression, and treatment response in CD. The aa genotype of ApaI appears to confer protection against adverse disease outcomes.

## 1. Introduction

Vitamin D is a fat-soluble vitamin that exerts pleiotropic immunomodulatory and anti-inflammatory effects, supporting gut barrier integrity and microbiome homeostasis [1]. These functions are largely mediated by the binding of biologically active vitamin D to its cognate receptor (vitamin D receptor, or VDR), a member of the nuclear receptor superfamily. VDR is nearly ubiquitously expressed in human tissues, particularly in intestinal epithelial cells, as well as in both innate and adaptive immune cells residing in the intestinal compartment [2]. The vitamin D/VDR complex promotes the prevalence of beneficial genera, enhances the secretion of antimicrobial peptides, regulates the expression of tight junctions, induces T-regulatory cell function, and suppresses Th-17-related inflammation [2,3]. Due to its multifaceted regulatory effects on the intestine, the disruption of vitamin D/VDR signaling has been implicated in the pathogenesis of inflammatory bowel diseases (IBD) [4]. Indeed, vitamin D deficiency is common among IBD patients and has been related to disease activity; however, it remains debatable whether reduced vitamin D levels represent a precipitator of the disease in genetically predisposed people or another consequence of the disease itself [5]. Moreover, intestinal VDR expression is compromised in IBD and demonstrates an inverse correlation with levels of inflammatory markers [6].

In addition to impaired bioavailability of its components, vitamin D/VDR downstream signaling can also be affected by genetic variations in the VDR gene. IBD has a strong genetic background, with over 200 identified risk-associated genes [7]. Many studies have explored the association between four common VDR single nucleotide polymorphisms (SNPs), namely, ApaI, BsmI, TaqI, and FokI, and the risk of developing IBD [8,9,10,11]. The VDR gene, located on chromosome 12 (12q13.11), comprises 11 exons [12,13]. The FokI polymorphism, found in exon 2, involves a C to T transition at a start codon position, encoding a structurally different protein. The f allele (T) produces a 427-amino acid protein, which is longer by three amino acids and shows an altered ability to induce the transcription of vitamin D-related genes compared to the isoform synthesized by the wild-type F allele (C). The ApaI and BsmI polymorphisms are located within intron 8, while the TaqI is found within exon 9, downstream of the coding region. These genetic variants do not generate a modified VDR transcript; however, they might influence VDR-mediated signaling by modulating alternative slicing and interacting with intronic regulatory elements [14]. Moreover, these polymorphisms are clustered near the 3’-untranslated region (UTR), potentially affecting mRNA stability [12,13]. Although numerous studies have shown a correlation between specific VDR SNPs and the risk of ulcerative colitis (UC) or Crohn’s disease (CD), meta-analyses have not consistently supported a clear pattern linking these genetic variations to IBD susceptibility [15,16,17]. Studies including patients from diverse ethnic backgrounds, variations in sample size, and differences in the classification between UC and CD may help to explain these discrepancies to some extent [18].

While much research has concentrated on the association between VDR polymorphisms and the risk of developing the disease, significantly less is known about how these genetic variations influence the phenotype and prognosis of IBD. In this study, we aim to examine the distribution of the ApaI, BsmI, TaqI, and FokI SNPs across different IBD phenotypes and behavioral patterns, as well as their relationship with clinical outcomes and complications.

## 2. Results

### 2.1. Patient Characteristics

The baseline characteristics of the patients are summarized in Table 1.

Demographic parameters were comparable between CD and UC patients apart from smoking status, which was more prevalent among CD patients (57.9% vs. 36.7%; *p* = 0.011). The distribution of ApaI, BsmI, TaqI, and FokI genotypes was similar between the CD and UC cohorts. Additionally, a higher percentage of CD patients had a history of IBD-related surgery compared to UC patients (36.8% vs. 8.8%; *p* < 0.001). The frequencies of VDR SNPs were consistent with Hardy–Weinberg equilibrium across both the CD and UC groups, as well as within the overall patient population, meaning that the observed genotype frequencies accurately reflect the general population without significant influence from selection, mutation, or non-random mating. Details of HWE analysis and codification of genotypes according to allelic variations are further described in Table 2.

### 2.2. The Relationship Between Disease Characteristics and VDR SNPs

The distribution of VDR SNPs among patients with different disease patterns is shown in Table 3.

Significant differences in distribution were identified for the ApaI polymorphism (*p* = 0.028) and disease behavior among CD patients. As shown in Figure 1A, patients with the AA genotype had an increased likelihood of developing penetrating disease (B3) compared to those carrying Aa/aa genotypes (36.7% vs. 8.7%; *p* = 0.012). Conversely, patients carrying the recessive allele of the ApaI SNP (Aa/aa) were more likely to present with stenotic disease (B2) (67.4% vs. 40%; *p* = 0.063) (Figure 1B). No significant differences in CD behavior were observed across patients with different BsmI, TaqI, or FokI variants.

Additionally, perianal complications in CD, including anal fistulae, fissures, and abscesses, did not appear to be influenced by VDR SNPs. However, the FokI polymorphism appeared to be associated with disease location in CD. Patients with the ff genotype more frequently experienced upper GI involvement (36.4% vs. 7.7% vs; *p* = 0.044) or colonic disease (90.9% vs. 50.8%; *p* = 0.038) in comparison to the FF/Ff genotypes (Figure 1C,D).

The extent and severity of UC were not influenced by ApaI, BsmI, TaqI, or FokI variants. The presence of extraintestinal manifestations (EIM), family history, and coexisting autoimmune disease were also not associated with VDR SNPs in both subgroups and the overall IBD cohort (Appendix A).

### 2.3. The Influence of VDR SNPs on Treatment Parameters in IBD

The majority of patients in our cohort had prior exposure to biologic therapy, with 81.6% of CD patients and 66.2% of UC patients currently receiving biologic treatment (*p* = 0.038). A subset of participants was also treated with immunomodulators, with 10.3% of UC patients and 9.2% of CD patients receiving these therapies. Additionally, 57.4% of UC patients were treated with 5-ASA. Most individuals exhibited a non-steroid-dependent or steroid-refractory course of IBD (73.7% for CD and 57.3% for UC; *p* = 0.052). Further treatment details are provided in Table 4.

No significant association was found between VDR SNPs and the need for biologic therapy in either the CD or UC cohorts.

Nevertheless, VDR SNPs appeared to influence the response to steroids as portrayed in Figure 2.

In CD, patients that were homozygous for the recessive ApaI allele (aa genotype) were at lower risk of developing steroid-dependent or steroid-refractory disease compared to those harboring the dominant allele (AA/Aa) (0% vs. 32.3%; *p* = 0.03). Additionally, CD patients with the FF genotype more frequently presented steroid-responsive disease relative to those with the Ff/ff genotypes (87.1% vs. 64.4%; *p* = 0.069). Across the entire cohort, carriers of the aa genotype were less likely to experience steroid-refractory or steroid-dependent disease compared to the AA/Aa genotypes (11.1% vs. 37.6%; *p* = 0.022).

### 2.4. VDR Polymorphisms and the Risk for IBD-Related Hospitalization

In Crohn’s disease, the risk of hospitalization related to IBD was significantly higher among patients homozygous for the dominant traits of the ApaI, BsmI, and TaqI polymorphisms compared to those with the corresponding recessive alleles.

Patients with the AA genotype exhibited the highest hospitalization rate (76.7%), followed by those with the Aa genotype (53.1%), and the aa genotype (28.6%). This pattern suggests an additive effect, where the A allele, especially in AA homozygotes, is associated with an increased risk of hospitalization. AA carriers had a significantly higher hospitalization rate than Aa/aa carriers (76.7% vs. 45.6%; *p* = 0.009) and aa carriers (76.7% vs. 28.6%; *p* = 0.006) (Figure 3A).

Similarly, individuals with the AA/Aa genotypes also showed a significantly greater risk of hospitalization compared to those with the aa genotype alone (64.5% vs. 28.6%; *p* = 0.018). The BB genotype was associated with higher hospitalization rates compared to Bb/bb genotypes (85.7% vs. 51.6%; *p* = 0.033) and the bb genotype alone (85.7% vs. 50%; *p* = 0.042). Similarly, IBD-related hospitalization was more frequent in TT genotype carriers compared to Tt/tt genotypes (82.4% vs. 50.8%; *p* = 0.026) and the tt genotype alone (82.4% vs. 48.2%; *p* = 0.03) (Figure 3B, C).

Binary logistic regression analysis further supported these findings, identifying disease behavior and the ApaI, BsmI, and TaqI polymorphisms as significant predictors of IBD-related hospitalization in the univariate analysis (Table 5).

Patients with the B3/B2 phenotypes demonstrated a significantly higher risk of hospitalization (OR = 5.34, 95% CI: 1.66–17.16; *p* = 0.005) compared to those with the B1 phenotype. The AA genotype increased the risk of IBD-related hospitalization by nearly fourfold (OR = 3.91, 95% CI: 1.4–10.91; *p* = 0.009), while the aa genotype appeared to confer protection (OR = 0.22, 95% CI: 0.06–0.78; *p* = 0.02). Additionally, the BB genotype was associated with a higher risk (OR = 5.63, 95% CI: 1.16–27.25; *p* = 0.032), as was the TT genotype (OR = 4.51, 95% CI: 1.17–17.36; *p* = 0.028), when compared to patients with the corresponding recessive alleles (Bb/bb and Tt/tt, respectively).

In the multivariate analysis, disease behavior and the ApaI polymorphism emerged as the only independent predictors of IBD-related hospitalization. Specifically, patients with B2/B3 disease behavior (aOR = 6.71, 95% CI: 1.85–24.35; *p* = 0.004) and the AA genotype (OR = 4.86, 95% CI: 1.54–15.32; *p* = 0.007) had a significantly higher risk of hospitalization.

To ensure the robustness of our findings, we tested alternative multivariate models (Table 5). The results were consistent across models, with disease behavior (Β3/B2 phenotype) and the ApaI polymorphism (AA genotype) repeatedly identified as independent predictors of this outcome. In model 2, all variables with a *p*-value ≤ 0.1 in the univariate analysis were included, and the results remained stable. In model 3, we introduced the recessive genotype (aa) as a predictor against the Aa/aa genotype. This model reaffirmed disease behavior as an independent risk factor for IBD-related hospitalization (aOR = 8.65, 95% CI: 2.28–32.88; *p* = 0.002), while the aa genotype demonstrated a protective effect (aOR = 0.17, 95% CI: 0.04–0.69; *p* = 0.013). In this model, the TT genotype also emerged as an independent risk factor for hospitalization (aOR = 4.79, 95% CI: 1.04–21.98; *p* = 0.044).

In patients with ulcerative colitis, none of the VDR polymorphisms were associated with hospitalization in the univariate analysis (Table 6).

### 2.5. VDR Polymorphisms and the Risk for IBD-Related Surgery

In Crohn’s disease, patients with the AA genotype exhibited the highest surgery rate (56.7%), followed by those with the Aa genotype (31.3%) and the aa genotype (7.1%), suggesting an additive effect, where the A allele, particularly in AA homozygotes, is associated with a progressively higher risk of surgery. The prevalence of IBD-related surgery was significantly higher in patients with the AA genotype compared to those with the Aa/aa genotypes (56.7% vs. 23.9%; *p* = 0.007) and the aa genotype alone (56.7% vs. 7.1%; *p* = 0.003). Furthermore, individuals with the AA/Aa genotypes demonstrated a higher surgery rate compared to those with the aa genotype (43.5% vs. 7.1%; *p* = 0.013) (Figure 3D). Similarly, IBD-related surgery was more frequent in patients harboring the TT genotype compared to participants with the Tt/tt genotypes (58.8% vs. 30.5%; *p* = 0.046) or the tt genotype alone (58.8% vs. 25.9%; *p* = 0.055).

In univariate analysis, factors associated with an increased risk of IBD-related surgery included age (OR = 1.03, 95% CI: 1.001–1.07; *p* = 0.046), B2/B3 disease behavior (OR = 14.8, 95% CI: 1.85–118.7; *p* = 0.011), the AA genotype of the ApaI SNP (OR = 4.16, 95% CI: 1.55–11.2; *p* = 0.005), and the TT genotype of the TaqI SNP (OR = 3.25, 95% CI: 1.07–9.91; *p* = 0.038) (Table 7).

Active smoking showed a trend towards increased risk (OR = 2.68, 95% CI: 0.99–7.28; *p* = 0.053), though this was not statistically significant. Conversely, the aa genotype was associated with a significantly reduced risk of this outcome (OR = 0.1, 95% CI: 0.12–0.81; *p* = 0.031).

In the multivariate analysis, including all variables with a *p*-value ≤ 0.05 from the univariate analysis (model 1), the ApaI polymorphism and disease behavior emerged as independent predictors of IBD-related surgery (Table 7). Specifically, the B2/B3 disease behavior (aOR = 20.75, 95% CI: 2.25–191.8; *p* = 0.008) and the AA genotype of the ApaI polymorphism (aOR = 6.1, 95% CI: 1.9–19.59; *p* = 0.002) were associated with a substantially higher risk. In model 2, which included all variables with a *p*-value ≤ 0.1 from univariate analysis, age (aOR = 1.04, 95% CI: 1.00–1.08; *p* = 0.033), B2/B3 phenotype (aOR = 23.4, 95% CI: 2.28–240.6; *p* = 0.008), and AA genotype (aOR = 4.22, 95% CI: 1.21–14.7; *p* = 0.024) were all significantly associated with increased risk of surgery. Model 3 includes all variables with a *p*-value ≤ 0.05, using the aa genotype as a comparison against the combined Aa/AA genotypes. In this model, age and B2/B3 disease behavior remained independently associated with increased risk, while the aa genotype was associated with a 45% reduction in the risk of surgery (aOR = 0.055, 95% CI: 0.006–0.56; *p* = 0.014).

In patients with ulcerative colitis, univariate analysis (Table 6) identified two factors associated with an increased risk of surgery: years since diagnosis (OR = 1.09, 95% CI: 1.01–1.17; *p* = 0.022) and the ff genotype of the FokI SNP (OR = 9.83, 95% CI: 1.26–76.84; *p* = 0.029). However, in multivariate regression analysis, only years since diagnosis remained an independent predictor of IBD-related surgery (aOR = 1.09, 95% CI: 1.01–1.17; *p* = 0.037).

### 2.6. Linkage Disequilibrium and Haplotype Analysis

Pairwise linkage disequilibrium (LD) analysis indicated a very strong linkage among certain SNPs in both UC and CD patients (Appendix A). Specifically, ApaI and BsmI (D′ > 0.9), BsmI and TaqI (0.9 > D′ > 0.8), as well as ApaI and TaqI (D′ > 0.9) exhibited high levels of LD, suggesting that these SNPs are likely co-inherited within the population. In UC, the FokI polymorphism showed mild LD with BsmI, TaqI, and ApaI (0.4 > D′ > 0.3). In CD, FokI exhibited mild to moderate LD with TaqI (D′ = 0.39) and BsmI (D′ = 0.46), and it exhibited moderate LD with ApaI (D′ = 0.6).

Haplotype analysis revealed that the abtF haplotype was associated with a reduced risk of surgery (OR = 0.29, 95% CI: 0.12–0.7; adjusted *p* = 0.01) and hospitalization (OR = 0.4, 95% CI: 0.19–0.83; adjusted *p* = 0.029) in patients with CD (Appendix A). These findings suggest that the abtF haplotype may have a protective effect against adverse outcomes in Crohn’s disease.

## 3. Discussion

Our findings suggest that VDR genetic variants play a role in shaping the phenotype and progression of IBD-related complications. Specifically, the ApaI polymorphism appears to influence the natural history of Crohn’s disease by driving the development of adverse outcomes. In particular, multivariate analysis revealed that the aa genotype of the ApaI SNP was independently associated with a reduced risk of IBD-related hospitalization and surgery in CD patients. In contrast, the AA genotype was linked to an increased risk of these adverse outcomes. Intriguingly, this association followed an additive pattern, with the aa genotype corresponding to the lowest risk, the Aa genotype to a moderate risk, and the AA genotype to the highest risk. The ApaI SNP also influenced disease behavior, with the AA genotype associated with penetrating CD in bivariate analysis. Additionally, the FokI SNP was found to affect disease localization, with the ff genotype predisposing patients to involvement of the colon and upper gastrointestinal tract. Furthermore, VDR SNPs modulated steroid treatment response, with patients carrying the A allele of the ApaI SNP (AA/Aa) developing steroid-refractory or steroid-dependent disease at higher rates across the entire cohort.

We have previously reviewed data from published meta-analyses on the relationship between VDR SNPs and susceptibility to IBD [18]. In a meta-analysis including nine studies, Xue et al. demonstrated that the recessive allele of the ApaI SNP conferred a protective effect against the development of CD. Moreover, the TaqI tt genotype increased the risk of CD in Europeans, while the FokI ff genotype predisposed Asian populations to UC [17]. Wang et al., in their meta-analysis including 3462 IBD patients, reported that the ApaI and TaqI SNPs were correlated with a greater likelihood of developing CD and UC, respectively, particularly among individuals of Caucasian ancestry [15]. Cho et al. demonstrated that the risk of developing CD and UC was higher in patients harboring the FokI f allele [16]. Although the results remain inconclusive and sometimes contradictory, VDR polymorphisms play a pivotal role in modulating vitamin D/VDR signaling, which influences immune responses, inflammation, and tissue repair mechanisms central to IBD pathogenesis. Notably, homozygosity for the FokI ancestral allele is associated with lower vitamin D levels in IBD, which, in turn, correlates with increased endoscopic activity of the disease [19]. Additionally, Xia et al. demonstrated that the combined presence of specific VDR polymorphisms (FokI, ApaI, TaqI) and vitamin D deficiency exert a synergistic impact on CD [20].

Recently, increasing attention has been given to the role of VDR genetic variants in shaping the phenotype of IBD. One study reported that individuals who are heterozygous for the BsmI and ApaI polymorphisms face an elevated risk of developing perianal disease in CD [21]. In contrast, our study found no significant association between these polymorphisms and the development of perianal CD. However, our study had a smaller sample size and may have been underpowered to detect such an association, whereas Cusato et al. analyzed a larger cohort of 206 CD patients, including 34 with perianal disease. Gisbert-Ferrándiz et al. demonstrated that the course of CD can be affected by the ApaI and BsmI SNPs, with specific genotypes driving variations in disease behavior [22]. For instance, patients carrying the recessive alleles of ApaI (Aa/aa) or BsmI (Bb/bb) SNPs were less likely to develop B3-type disease. Moreover, the AA/BB/tt haplotype was identified as a predictor of penetrating disease, while the aa genotype appeared protective against perianal fistula formation. Consistent with these findings, our data indicate that patients carrying the ApaI recessive allele (Aa/aa) experience lower rates of penetrating CD. Additionally, while Gisbert-Ferrándiz et al. associated the bb genotype with a reduced risk of surgery, our analysis linked the opposite genotype, BB, to an increased risk of hospitalization in univariate analysis [22]. Further supporting the role of VDR variants, a prior study revealed that patients with the TaqI TT genotype exhibited reduced VDR expression in peripheral blood mononuclear cells, leading to upregulation of pro-inflammatory IL1β mRNA levels and enhanced activation of lymphocytic adhesion molecules [23]. This genotype was also associated with a higher risk of developing a B3-penetrating phenotype and an increased likelihood of undergoing surgery [23]. Our results align with the above data, as we identified the TT genotype as an independent predictor of IBD-related hospitalization in the multivariate analysis and IBD-related surgery in the univariate analysis. These findings offer a molecular insight into how variations in VDR/vitamin D signaling, driven by genetic differences, may contribute to distinct disease patterns.

The gut harbors a diverse consortium of microorganisms that regulate mucosal immune responses and support barrier integrity. Vitamin D/VDR signaling plays a crucial role in maintaining microbiome homeostasis, as higher vitamin D levels contribute to an abundance of symbiotic bacteria [24]. In contrast, vitamin D or VDR deficiency induces dysbiosis and increases susceptibility to intestinal injury, as demonstrated in animal models [25]. IBD is characterized by a profound dysregulation of intestinal microflora, marked by the overgrowth of harmful species and depletion of beneficial genera, leading to an aberrant influx of toxins and excessive inflammatory activation. Genetic polymorphisms in genes related to vitamin D signaling can further alter the microbiome, exacerbating the dysregulation observed in IBD.

A recent study illustrated how the TaqI polymorphism affects microbiota composition using 16S rRNA sequencing [26]. Analysis of alpha- and beta-diversity revealed significant differences in phylogenetic diversity across TaqI genotypes (CC, TT, TC), supporting the idea that these variations arise from the presence of distinct bacterial species rather than differences in their abundance. The tt (TT) genotype exhibited higher phylogenetic diversity, hosting increased levels of *Parabacteroides* and *Butyricimonas*, which modulate complex carbohydrate catabolism and short-chain fatty acid synthesis [26]. Another study demonstrated that the TaqI polymorphism is associated with alterations in gut microbiota composition, potentially increasing the likelihood of harboring the opportunistic pathogen *Solobacterium* [27]. These genetic variations influence not only the microbiota but also play a critical role in the inflammatory mechanisms underlying IBD, suggesting a pathway through which specific SNPs may drive disease progression. In obesity, a risk haplotype was associated with increased expression of inflammasome components, upregulation of proinflammatory cytokines, and reduced VDR expression. Additionally, individuals with this haplotype showed higher plasma LPS levels, indicating compromised gut permeability that may facilitate microbial translocation [28]. These findings suggest that polymorphisms affecting the vitamin D/VDR pathway could contribute to inflammation driven by a “leaky” gut and subsequent endotoxemia.

Genetic polymorphisms in the VDR gene are associated with disease outcomes and treatment responses in various oncologic, infectious, and autoimmune conditions [29,30,31]. For example, the FokI and TaqI SNPs influence the time to sputum culture and auramine stain conversions during tuberculosis treatment [32]. In addition, the FokI recessive allele predicts treatment failure in patients with chronic hepatitis C undergoing therapy with pegylated interferon and ribavirin [33]. In IBD, VDR mRNA expression—regulated by VDR genetic polymorphisms—plays a pivotal role in shaping the response to biologic therapy. Notably, an artificial intelligence-based algorithm incorporating VDR predicts primary non-response to infliximab in UC [34]. Furthermore, certain TaqI and FokI polymorphisms yield superior responses to vitamin D supplementation, emphasizing the role of VDR genetic variants in modulating treatment outcomes [35]. Our data support that patients carrying the dominant ApaI (AA/Aa) or the recessive FokI (Ff/ff) alleles are more susceptible to steroid-refractory or steroid-dependent CD. Interestingly, the FokI f allele is also associated with glucocorticoid-resistant asthma [36].

We acknowledge several limitations of our study. First, it primarily involved patients of Caucasian ancestry from a single referral center, which may limit the generalizability of our findings. However, the distribution of VDR SNPs in our sample was consistent with that found in other studies, suggesting that our sample is representative of the broader IBD population [15,16,17]. Additionally, while the sample size was relatively small, it was determined through power calculations based on previous studies, ensuring that it was adequate for the analysis. To address the risk of type I errors, we applied Fisher’s exact test for 2 × 2 contingency table analysis, a more conservative method for smaller samples. We recommend that future multicenter studies with more diverse populations be conducted to validate and expand upon our results, enhancing their generalizability and applicability.

Overall, our data underscore the substantial genetic contribution to IBD, demonstrating that VDR SNPs affect the clinical presentation and progression of the disease. In fact, the ApaI SNP emerged as an independent predictor of a definitive and robust endpoint: IBD-related surgery. The association followed an incremental pattern, where each additional copy of the risk allele (from aa to Aa to AA) corresponded to a progressively higher risk of adverse outcomes. Furthermore, the AA genotype of ApaI and the TT genotype of TaqI SNPs were identified as independent predictors of another important, though less straightforward, endpoint, i.e., IBD-related hospitalization. In the bivariate analysis, the ApaI SNP was associated with CD behavior, with the dominant allele (A) favoring penetrating disease, while the FokI SNP affected disease location, with the ff genotype predisposing to involvement of the colon and upper GI tract. The ApaI aa genotype emerged as a marker for steroid-responsive disease. Further studies are required to clarify the causal relationship between VDR genetic variants and IBD phenotype. Establishing VDR polymorphisms as reliable biomarkers could help predict treatment response and guide personalized therapy. Additionally, this research could enable the stratification of patients based on their genetic risk for complications, thereby improving management and clinical outcomes.

## 4. Materials and Methods

### 4.1. Study Design and Participants

For this observational exploratory study, a total of 144 patients with a confirmed diagnosis of IBD were consecutively enrolled from a tertiary referral center in Southwestern Greece (Division of Gastroenterology, Patras University Hospital) between September 2021 and September 2023. Of these patients, 76 were diagnosed with CD and 68 with UC based on solid clinical, laboratory, endoscopic, and histologic findings. All patients were treated according to the European Crohn’s and Colitis Organization’s (ECCO) guidelines [37,38]. Exclusion criteria included: age <18 years; indeterminate colitis; ischemic colitis; radiation colitis; microscopic colitis; diverticular disease-associated colitis; significant concurrent, uncontrolled medical conditions (e.g., active malignancy); hereditary colorectal cancer syndrome.

This study conforms to the ethical principles outlined in the Declaration of Helsinki for medical research involving human subjects. All participants provided written informed consent prior to their involvement in the study. The study protocol received approval from both the Scientific Review Board (Reference: 303/20-07-2021) and the Ethics Committee (Reference: 237/14-05-2021) of Patras University.

### 4.2. Clinical, Laboratory, and Endoscopic Evaluation

A comprehensive medical history, including epidemiological and demographic characteristics, was collected from all patients at the time of enrollment. Blood samples were obtained for VDR genotyping and routine blood tests. The Crohn’s Disease Activity Index (CDAI) and Harvey-Bradshaw Index (HBI) were calculated for patients with Crohn’s disease [39,40], while the Mayo Score was used for those with UC [41]. For patients who underwent endoscopy at recruitment, endoscopic activity scores were recorded using the Simple Endoscopic Score for Crohn’s Disease (SES-CD) and the Mayo Endoscopic Score for UC [41,42]. For patients who did not have an endoscopy at inclusion, these indices were recorded using data from six months prior to or following enrollment during the follow-up examination.

### 4.3. Disease Phenotype

The Montreal Classification was used to categorize the clinical characteristics of IBD [43]. For both CD and UC, age at diagnosis was classified as A1, under 16 years; A2, 17–40 years; and A3, over 40 years. For CD, location was defined as follows: L1, disease confined to the ileum; L2, disease limited to the colon; L3, disease involving both ileum and colon; and L4-modifier, disease affecting the upper gastrointestinal tract, which may be combined with L1-L3. CD behavior was categorized into: B1, non-stricturing, non-penetrating; B2, stricturing; B3, penetrating; and the perianal disease modifier (p), which can accompany B1-B3 phenotypes. For UC, the extent of disease was classified as E1 for ulcerative proctitis (limited to the rectum), E2 for left-sided colitis (extending up to the splenic flexure), and E3 for extensive colitis (extending proximal to the splenic flexure). UC severity was categorized into S0 for clinical remission; S1, mild disease (fewer than four stools daily, with or without blood, and no systemic toxicity); S2, moderate disease (more than four stools daily, with minimal systemic toxicity); and S3, severe disease (more than six bloody stools daily with systemic toxicity).

### 4.4. Outcomes and Definitions

The medical records of the patients were reviewed, and treatment history data were extracted. Adverse outcomes, such as IBD-related surgery or IBD-related hospitalization, were documented. Only hospitalizations related to disease symptoms or flare-ups were considered, excluding those for procedural needs such as colonoscopies or intravenous infusions. Hospitalizations were identified as disease-related if they involved IBD flare-ups or symptoms and required more than one day of hospitalization. IBD-related surgeries were documented based on the presence of surgical records or operative reports. The ECCO definitions were used to characterize steroid-refractory and steroid-dependent disease. Steroid-dependent disease refers to patients who: (i) were unable to taper steroids below the equivalent of 10 mg/day of prednisolone (or 3 mg/day of budesonide) within three months of starting treatment without recurrent active disease; (ii) experienced a relapse within 3 months of discontinuing steroids. Steroid-refractory disease was defined as persistent active disease despite treatment with prednisolone up to 1 mg/kg/day over a period of 4 weeks [44,45].

### 4.5. VDR Genotyping

Genotyping of VDR polymorphisms was conducted using TaqMan SNP genotyping assays (Applied Biosystems, Foster City, CA, USA). PCR reactions were performed in MicroAmp Fast Optical 96 well Reaction Plates (Applied Biosystems, Foster City, CA, USA) on a Step One Plus real-time PCR system (Applied Biosystems, Foster City, CA, USA). Predesigned TaqMan SNP genotyping assays (Applied Biosystems, Foster City, CA, USA) were used to target the following VDR SNPs: rs731236 (TaqI), rs1544410 (BsmI), rs7975232 (ApaI), and rs2228570 (FokI). Each plate contained two wells with healthy non-template controls. The PCR cycling conditions for DNA amplification were as follows: 95 °C for 10 min, followed by 40 cycles of 95 °C for 15 s and 60 °C for 1 min.

### 4.6. Sample Size Consideration

A priori power analysis was conducted to determine the required sample size for a logistic regression model examining the association between a genetic variant and outcome risk. Based on previous research, we assumed an odds ratio of 3.5, a 40% baseline risk of the outcome, a significance level of 0.05, and a desired power of 80% [23]. The analysis indicated that a sample size of 68 participants per IBD sub-cohort (UC and CD) would be necessary. This calculation assumed a binomial distribution for the predictor with an equal probability for each category. G*Power software version 3.1.9.4 was utilized for sample size estimation.

### 4.7. Statistics

Continuous variables were reported as medians with interquartile ranges (IQR), while categorical data were presented as counts with their respective percentages. Comparisons between continuous variables were performed using the Mann–Whitney U test or the Student’s *t*-test when data met normality criteria as assessed by the Shapiro–Wilk test. For categorical variables, contingency tables were created for the three genotypes of each SNP and analyzed using the Pearson’s χ^2^ test (3 × 2 and 3 × 3 tables). When significant results were obtained, further analysis was performed using Fisher’s exact test (2 × 2 tables) to enhance precision, comparing the genotype of interest against the combined group of the other two genotypes for the SNP. Bonferroni correction for multiple comparisons was applied. Hardy–Weinberg Equilibrium (HWE) was assessed for each SNP in the total population and subgroups using Pearson’s χ^2^ test with one degree of freedom, comparing observed and expected genotype frequencies. Pairwise linkage disequilibrium (LD) analysis between VDR variants and haplotype analysis were conducted using the SHEsis software platform (http://shesisplus.bio-x.cn/SHEsis.html, accessed on 18 February 2025) [46,47]. In the haplotype analysis, *p*-values were adjusted using the step-up procedure by Benjamini and Hochberg to control the false discovery rate (FDR) [48]. Binary logistic regression was used to assess variables associated with disease-related complications and to calculate the odds ratio (OR) along with 95% confidence intervals (CIs). A backward stepwise approach was utilized for the multivariate analysis, incorporating all variables with a *p* ≤ 0.05 from the univariate analysis. Additional multivariate models were implemented to validate the results, using different *p*-value (*p* ≤ 0.1) thresholds for variable inclusion. Furthermore, for variables with multiple categories (e.g. VDR SNPs), we tested different reference groups in the multivariate models when they were significant in the univariate analysis. The threshold for statistical significance was set at *p* = 0.05. Statistical analysis was performed using the statistical software package IBM SPSS version 26 and GraphPad Prism version 9.

## Figures and Tables

**Figure 1 ijms-26-01848-f001:**
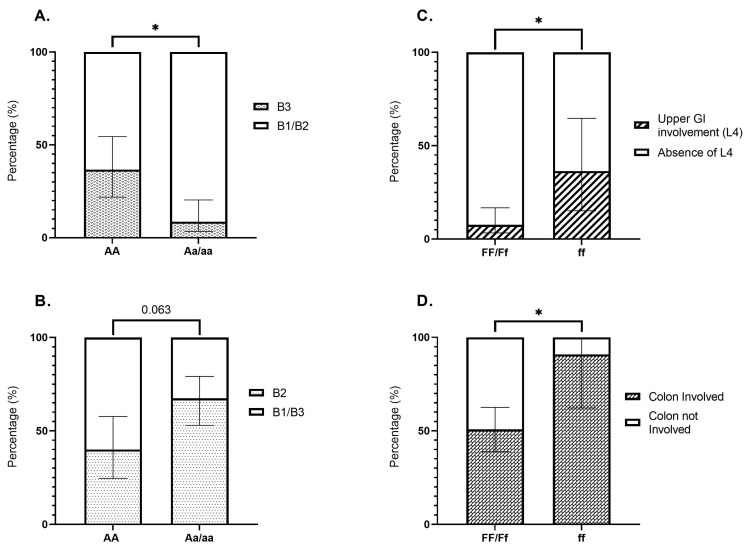
(**A**) Proportion of penetrating Crohn’s disease (B3) in patients with AA vs. Aa/aa genotypes (36.7% vs. 8.7%; *p* = 0.012). (**B**) Proportion of stenotic Crohn’s disease (B2) in patients with AA vs. Aa/aa genotypes (40% vs. 67.4%; *p* = 0.063). (**C**) Proportion of upper GI involvement (L4) in Crohn’s disease patients with FF/Ff vs. ff genotypes (7.7% vs. 36.4%; *p* = 0.044). (**D**) Proportion of colonic involvement in Crohn’s disease patients with FF/Ff vs. ff genotypes (50.8% vs. 90.9%; *p* = 0.038). Bonferroni correction for multiple comparisons was applied. * *p* < 0.05.

**Figure 2 ijms-26-01848-f002:**
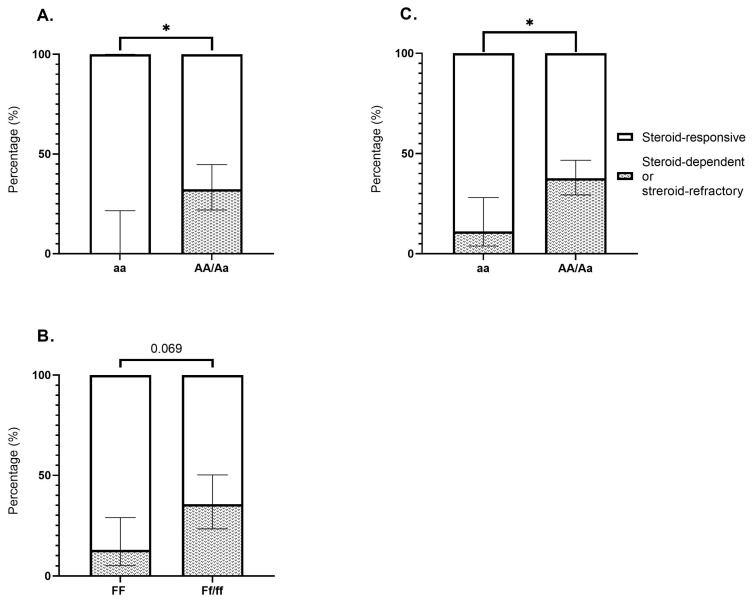
(**A**) Proportion of steroid-refractory or steroid-dependent disease in Crohn’s disease patients with aa vs. AA/Aa genotypes (0% vs. 32.3%; *p* = 0.03). (**B**) Proportion of steroid-refractory or steroid-dependent disease in Crohn’s disease patients with FF vs. Ff/ff genotypes (12.9% vs. 35.6%; *p* = 0.069). (**C**) Proportion of steroid-refractory or steroid-dependent disease in IBD patients with aa vs. AA/Aa genotypes (11.1% vs. 37.6%; *p* = 0.022). Bonferroni correction for multiple comparisons was applied. * *p* < 0.05.

**Figure 3 ijms-26-01848-f003:**
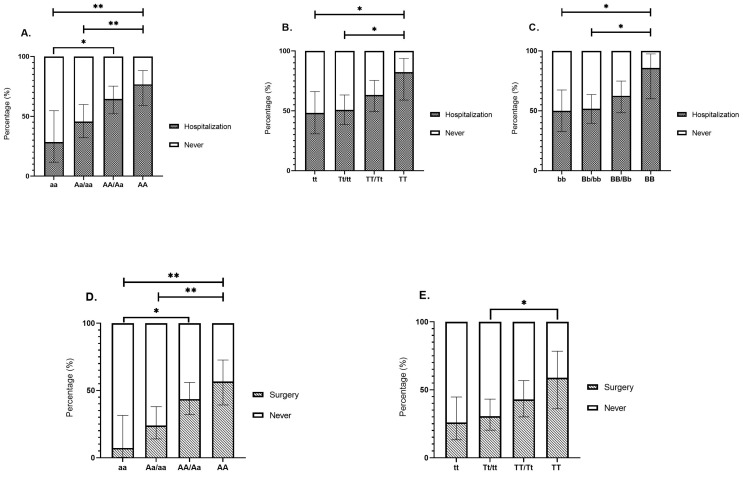
Association of VDR SNPs with IBD-related hospitalization (**A**–**C**) and surgery (**D**,**E**) rates in Crohn’s disease. (**A**) Carriers of the AA genotype of the ApaI SNP had a significantly higher hospitalization rate compared to individuals with the combined Aa/aa genotypes (76.7% vs. 45.6%; *p* = 0.009) and those with the aa genotype alone (76.7% vs. 28.6%; *p* = 0.006). Additionally, individuals with either the AA or Aa genotype showed a significantly greater risk of hospitalization compared to those with the aa genotype alone (64.5% vs. 28.6%; *p* = 0.018). This trend suggests an additive effect, with the presence of the A allele—particularly in AA homozygotes—being associated with an increased risk of hospitalization. (**B**) IBD-related hospitalization was more frequent in TT genotype carriers compared to Tt/tt genotypes (82.4% vs. 50.8%; *p* = 0.026) and the tt genotype alone (82.4% vs. 48.2%; *p* = 0.03). (**C**) The BB genotype was associated with higher hospitalization rates compared to the Bb/bb genotypes (85.7% vs. 51.6%; *p* = 0.033) and the bb genotype alone (85.7% vs. 50%; *p* = 0.042). (**D**) The prevalence of IBD-related surgery was significantly higher in patients with the AA genotype compared to those with the Aa/aa genotypes (56.7% vs. 23.9%; *p* = 0.007) and the aa genotype alone (56.7% vs. 7.1%; *p* = 0.003). Furthermore, individuals with the AA/Aa genotypes exhibited a higher surgery rate than those with the aa genotype (43.5% vs. 7.1%; *p* = 0.013), suggesting an additive effect of the A allele on the risk of surgery. (**E**) Individuals with the TT genotype had a higher surgery rate than those with the Tt/tt genotypes (58.8% vs. 30.5%; *p* = 0.046). * *p* < 0.05, ** *p* < 0.01.

**Table 1 ijms-26-01848-t001:** Patient characteristics.

	Crohn’s Disease (*n* = 76)	Ulcerative Colitis (*n* = 68)	Sig.
Age (years)	43 (34–54)	46 (35.5–60)	0.339
Years from diagnosis	9 (3–16)	10 (4.5–18)	0.65
Gender (female)	34 (44.7%)	34 (50%)	0.527
Smoker (current or former)	44 (57.9%)	25 (36.7%)	0.011 *
Family history of IBD	5 (6.6%)	3 (4.4%)	0.571
Extraintestinal manifestations	21 (2 7.6%)	20 (29.4%)	0.813
Coexisting auto-immune conditions	7 (9.2%)	7 (10.3%)	0.827
Body mass index (kg/m^2^)	24.3 (21.7–28.1)	24.8 (22.4–27.5)	0565
Hemoglobin (g/dL)	13.7 (12.7–14.5)	13.3 (11.5–14.6)	0.3
WBC (cells × 10^9^/L)	6.85 (5.8–9.31)	6.95 (5.06–9.3)	0.463
Platelets (cells × 10^9^/L)	297 (235–364)	277 (224–367)	0.545
CRP (mg/L)	6.3 (2–17)	3.5 (2–12)	0.246
ESR (mm/h)	22 (9.5–32)	20 (6.5–33)	0.627
Albumin (g/dL)	4.2 (4–4.5)	4.1 (3.7–4.5)	0.367
CDAI	61 (37–111)		n/a
HBI	2 (1–3)		n/a
SES-CD	3 (2–6)		n/a
Mayo Endoscopic Score		0 (0–2)	n/a
Full Mayo Score		3 (0–7)	n/a
IBD-related hospitalization	44 (57.9%)	35 (51.4%)	0.438
IBD-related surgery	28 (36.8%)	6 (8.8%)	<0.001 *
Follow-up (years)	9 (3–16)	10 (4.5–18)	0.65
Montreal Classification			
Age at diagnosis			
≤16 (A1)	7 (9.2%)	7 (10.3%)	0.942
17–40 (A2)	46 (60.5%)	42 (61.8%)	
>40 (A3)	23 (30.3%)	19 (27.9%)	
Location (CD)			
Terminal Ileum (L1)	33 (43.4%)		n/a
Colon (L2)	9 (11.8%)		
Ileo-colonic (L3)	34 (44.7%)		
Upper GI modifier (L4)	9 (11.8%)		
Behavior (CD)			
Non-stricturing non-penetrating (B1)	18 (23.7%)		n/a
Stricturing (B2)	43 (56.6%)		
Penetrating (B3)	15 (19.7%)		
Perianal disease modifier (p)	21 (27.6%)		
Extent (UC)			
Ulcerative proctitis (E1)		8 (11.8%)	n/a
Left-sided UC (E2)		32 (47%)	
Extensive UC (E3)		28 (41.2%)	
Severity (UC)			
Clinical remission (S0)		9 (13.2%)	n/a
Mild (S1)		29 (42.7%)	
Moderate (S2)		21 (30.9%)	
Severe (S3)		9 (13.2%)	

WBC: white blood cells; CRP: C-reactive protein; ESR: erythrocyte sedimentation rate; CDAI: Crohn’s disease activity index; HBI: Harvey-Bradshaw index; SES-CD: simple endoscopic score for Crohn’s disease; n/a: not applicable. * *p* < 0.05.

**Table 2 ijms-26-01848-t002:** Distribution and Hardy–Weinberg equilibrium (HWE) analysis of ApaI, BsmI, TaqI, and FokI polymorphisms in Crohn’s disease and ulcerative colitis.

SNP Name (SNP ID)	Codification of Nucleotide Change	Genotypes	Crohn’s Disease	Testing for HWE in CD χ^2^ (Sig.)	Ulcerative Colitis	Testing for HWE in UC χ^2^ (Sig.)	Test for SNPs Distribution CD vs. UC χ^2^ (Sig.)
ApaI (rs2228570)	A > C A > a	aa	14 (18.4%)	1.073 (*p* > 0.05)	13 (19.1%)	0.01 (*p* > 0.05)	0.01 (*p* > 0.05)
		Aa	32 (42.1%)		33 (48.5%)		
		AA	30 (39.5%)		22 (32.4%)		
BsmI (rs1544410)	A > G B > b	bb	28 (36.9%)	0.414 (*p* > 0.05)	26 (38.2%)	2.14 (*p* > 0.05)	1.98 (*p* > 0.05)
		Bb	34 (44.7%)		35 (51.5%)		
		BB	14 (18.4%)		7 (10.3%)		
TaqI (rs731236)	C > T T > t	tt	27 (35.5%)	1.555 (*p* > 0.05)	25 (36.8%)	0.233 (*p* > 0.05)	2.16 (*p* > 0.05)
		Tt	32 (42.1%)		34 (50%)		
		TT	17 (22.4%)		9 (13.2%)		
FokI (rs2228570)	C > T F > f	ff	11 (14.5%)	0.114 (*p* > 0.05)	5 (7.4%)	0.099 (*p* > 0.05)	3.9 (*p* > 0.05)
		Ff	34 (44.7%)		25 (36.7%)		
		FF	31 (40.8%)		38 (55.9%)		

SNP: single nucleotide polymorphism.

**Table 3 ijms-26-01848-t003:** Distribution of ApaI, BsmI, TaqI, and FokI polymorphisms across Crohn’s disease and ulcerative colitis phenotypes. * *p* < 0.05.

		Behavior (CD)				Perianal Disease (CD)			Upper GI Involvement (CD)			Colon Involvement (CD)			Extent (UC)				Severity (UC)		
		B1	B2	B3	Sig.	Yes	No	Sig.	Yes	No	Sig.	Yes	No	Sig.	E1	E2	E3	Sig.	S0/S1	S2/S3	Sig.
ApaI	aa	2	11	1	0.028 *	4	10	0.904	2	12	0.909	10	4	0.463	1	6	6	0.813	8	5	0.487
	Aa	9	20	3		8	24		4	28		17	15		3	17	13		16	17	
	AA	7	12	11		9	21		3	27		16	14		4	9	9		14	8	
BsmI	bb	7	17	4	0.532	9	19	0.749	3	25	0.944	17	11	0.827	4	11	11	0.918	16	10	0.652
	Bb	9	19	6		8	26		4	30		18	16		3	18	14		19	16	
	BB	2	7	5		4	10		2	12		8	6		1	3	3		3	4	
TaqI	tt	6	17	4	0.796	9	18	0.525	3	24	0.986	16	11	0.874	4	11	10	0.634	16	9	0.564
	Tt	8	18	6		9	23		4	28		17	15		2	17	15		17	17	
	TT	4	8	5		3	14		2	15		10	7		2	4	3		5	4	
FokI	ff	2	7	2	0.876	3	8	0.138	4	7	0.024 *	10	1	0.043 *	1	3	1	0.544	3	2	0.539
	Ff	9	17	8		13	21		3	31		18	16		4	9	12		16	9	
	FF	7	19	5		5	26		2	29		15	16		3	20	15		19	19	

**Table 4 ijms-26-01848-t004:** Analysis of participants’ treatment history and current therapies. * *p* < 0.05.

	Crohn’s Disease (*n* = 76)	Ulcerative Colitis (*n* = 68)	Sig.
5-ASA	10 (13.2%)	39 (57.4%)	<0.001 *
Immunomodulator experienced	18 (23.7%)	16 (23.5%)	0.99
Current immunomodulator	7 (9.2%)	7 (10.3%)	0.99
Azathioprine	4 (5.3%)	4 (5.9%)	
Methotrexate	3 (3.9%)	3 (4.4%)	
Biologic experienced	63 (82.9%)	46 (67.6%)	0.051
Current biologic	62 (81.6%)	45 (66.2%)	0.038 *
Infliximab	28 (36.8%)	24 (35.3%)	
Adalimumab	15 (19.7%)	3 (4.4%)	
Golinumab	0 (0%)	1 (1.5%)	
Anti-IL12/23 agent	13 (17.1%)	4 (5.9%)	
Vedolizumab	6 (7.9%)	13 (19.1%)	
Steroid-refractory or steroid-dependent disease	20 (26.3%)	29 (42.6%)	0.052

**Table 5 ijms-26-01848-t005:** Univariate and multivariate analyses of variables associated with IBD-related hospitalization in Crohn’s disease patients using binary logistic regression (based on a backward stepwise approach).

	Univariate Analysis	OR (95% CI)	Multivariate Analysis Model 1	aOR (95% CI)	Model 2	aOR (95% CI)	Model 3	aOR (95% CI)
Age	0.328	1.015 (0.98–1.05)						
Years from diagnosis	0.095	1.05 (0.99–1.1)			0.173	1.04 (0.98–1.11)		
Gender, male	0.749	1.16 (0.47–2.9)						
Active smoker	0.236	1.77 (0.69–4.54)						
Body mass index	0.696	1.02 (0.93–1.11)						
Behavior			0.004 **	6.71 (1.85–24.35)	0.004 **	6.71 (1.85–24.35)	0.002 **	8.65 (2.28–32.88)
B2 vs. B1	0.024 *	3.98 (1.2–13.19)						
B3 vs. B1	0.002 **	16.9 (2.76–103.38)						
B2/B3 vs. B1	0.005 **	5.34 (1.66–17.16)						
Perianal disease	0.662	1.26 (0.45–3.52)						
Treatment with biologics	0.127	2.6 (0.76–8.87)						
CDAI	0.572	1.002 (0.99–1.01)						
HBI	0.996	1.001 (0.82–1.22)						
SES-CD	0.342	1.12 (0.89–1.4)						
ApaI			0.007 **	4.86 (1.54–15.32)	0.007 **	4.86 (1.54–15.32)	0.013 *	0.17 (0.04–0.69)
AA vs. Aa/aa	0.009 *	3.91 (1.4–10.91)						
AA vs. aa	0.004 *	8.21 (1.96–34.51)						
aa vs. AA/Aa	0.02 *	0.22 (0.06–0.78)						
BsmI			0.665	1.61 (0.19–13.79)	0.781	1.36 (0.16–11.81)	0.798	1.33 (0.15–11.48)
BB vs. Bb/bb	0.032 *	5.63 (1.16–27.25)						
BB vs. bb	0.036 *	6 (1.12–31.88)						
bb vs. BB/Bb	0.289	0.6 (0.23–1.54)						
TaqI			0.282	2.61 (0.46–14.98)	0.403	2.12 (0.36–12.3)	0.044 *	4.79 (1.04–21.98)
TT vs. Tt/tt	0.028 *	4.51 (1.17–17.36)						
TT vs. tt	0.03 *	5.03 (1.17–21.59)						
tt vs. TT/Tt	0.204	0.54 (0.21–1.4)						
FokI								
FF vs. Ff/ff	0.654	0.81 (0.32–2.04)						
FF vs. ff	0.777	0.82 (0.2–3.33)						
ff vs. FF/Ff	0.677	1.32 (0.35–4.97)						

Model 1 includes all variables with *p*-value ≤ 0.05: behavior (B2/B3 vs. B1), ApaI (AA vs. Aa/aa), BsmI (BB vs. Bb/bb), and TaqI (TT vs. Tt/tt). Model 2 includes all variables with a *p*-value ≤ 0.1: behavior (B2/B3 vs. B1), ApaI (AA vs. Aa/aa), BsmI (BB vs. Bb/bb), TaqI (TT vs. Tt/tt), and years from diagnosis. Model 3 included all variables with a *p*-value ≤ 0.05, using the ApaI recessive genotype (aa) as a predictor: behavior (B2/B3 vs. B1), ApaI (aa vs. AA/Aa), BsmI (BB vs. Bb/bb), and TaqI (TT vs. Tt/tt). CDAI: Crohn’s disease activity index; HBI: Harvey-Bradshaw index; SES-CD: simple endoscopic score for Crohn’s disease. * *p* < 0.05, ** *p* < 0.01.

**Table 6 ijms-26-01848-t006:** Univariate and multivariate analyses of variables associated with IBD-related hospitalization and surgery in ulcerative colitis patients using binary logistic regression (based on a backward stepwise approach).

	IBD-Related Hospitalization				IBD-Related Surgery			
	Univariate Analysis	OR (95% CI)	Multivariate Analysis ^†^	aOR (95% CI)	Univariate Analysis	OR (95% CI)	Multivariate Analysis ^‡^	aOR (95% CI)
Age	0.682	0.994 (0.96–1.02)			0.1	1.05 (0.99–1.11)		
Years from diagnosis	0.917	0.998 (0.95–1.05)			0.022 *	1.09 (1.01–1.17)	0.037 *	1.09 (1.01–1.17)
Gender, male	0.467	1.43 (0.55–3.7)			0.401	0.47 (0.08–2.75)		
Active smoker	0.849	1.13 (0.34–3.78)			0.873	0.83 (0.09–7.81)		
Body mass index	0.102	1.16 (0.97–1.38)			0.428	0.85 (0.56–1.28)		
Extent (Ε3 vs. E2/E1)	0.08	2.44 (0.9–6.59)	0.369	1.67 (0.55–5.09)	0.647	1.48 (0.28–7.93)		
Severity (S2/S3 vs. S0/S1)	0.085	2.38 (0.89–6.35)	0.134	2.26 (0.78–6.58)	0.26	2.77 (0.47–16.27)		
Treatment with biologics	0.002 **	6.38 (1.98–20.48)	0.002 **	6.38 (1.98–20.48)	0.404	2.56 (0.28–23.36)		
Full Mayo Score	0.361	1.07 (0.93–1.24)			0.291	1.15 (0.89–1.48)		
Mayo Endoscopic Score	0.698	1.09 (0.7–1.72)			0.567	1.24 (0.59–2.59)		
ApaI			0.272	0.52 (0.17–1.66)				
AA vs. Aa/aa	0.089	0.402 (0.14–1.15)			0.404	0.39 (0.04–3.56)		
AA vs. aa	0.154	0.36 (0.09–1.47)			0.295	0.26 (0.02–3.22)		
aa vs. AA/Aa	0.422	1.66 (0.48–5.71)			0.365	2.32 (0.38–14.28)		
BsmI								
BB vs. Bb/bb	0.632	0.68 (0.14–3.3)			0.596	1.87 (0.19–18.73)		
BB vs. bb	0.487	0.55 (0.1–2.97)			0.596	2 (0.15–25.92)		
bb vs. BB/Bb	0.42	1.5 (0.56–4.02)			0.796	0.79 (0.13–4.66)		
TaqI								
TT vs. Tt/tt	0.793	1.21 (0.3–4.95)			0.796	1.35 (0.14–13.09)		
TT vs. tt	0.816	0.83 (0.18–3.88)			0.943	0.92 (0.08–10.14)		
tt vs. TT/Tt	0.285	1.73 (0.64–4.69)			0.486	1.82 (0.34–9.78)		
FokI							0.058	9.95 (0.93–106.79)
FF vs. Ff/ff	0.785	0.88 (0.34–2.28)			0.26	0.36 (0.06–2.12)		
FF vs. ff	0.676	0.68 (0.1–4.45)			0.033 *	0.08 (0.01–0.82)		
ff vs. FF/Ff	0.693	1.45 (0.23–9.3)			0.029 *	9.83 (1.26–76.84)		

^†^ Multivariate analysis (back included all variables with a *p*-value ≤ 0.1), as only one variable (treatment with biologics) had a *p*-value ≤ 0.05 in univariate analysis. This model included: disease extent (Ε3 vs. E2/E1), disease severity (S2/S3 vs. S0/S1), treatment with biologics, and ApaI (AA vs. Aa/aa). ^‡^ Multivariate analysis included all variables with a *p*-value ≤ 0.05: years from diagnosis and FokI (ff vs. FF/Ff). The only factor significantly linked to an increased risk of hospitalization was the need for treatment with biologic agents, with an odds ratio of 6.38 (95% CI: 1.98–20.48; *p* = 0.002). * *p* < 0.05, ** *p* < 0.01.

**Table 7 ijms-26-01848-t007:** Univariate and multivariate analysis of variables associated with IBD-related surgery in Crohn’s disease patients using binary logistic regression (based on a backward stepwise approach).

	Univariate Analysis	OR (95% CI)	Multivariate Analysis Model 1	aOR (95% CI)	Model 2	aOR (95% CI)	Model 3	aOR (95% CI)
Age	0.046 *	1.03 (1.001–1.07)	0.051	1.04 (1–1.08)	0.033 *	1.05 (1.004–1.09)	0.04 *	1.05 (1.002–1.09)
Years from diagnosis	0.13	1.04 (0.98–1.09)						
Gender, male	0.239	0.57 (0.22–1.46)						
Active smoker	0.053	2.68 (0.99–7.28)			0.106	2.97 (0.8–11.1)		
Body mass index	0.901	0.99 (0.91–1.08)						
Behavior			0.008 **	20.75 (2.25–191.8)	0.008 **	23.4 (2.28–240.6)	0.004 **	23.85 (2.69–211.6)
B2 vs. B1	0.02 *	12.24 (1.49–100.5)						
B3 vs. B1	0.005 *	25.5 (2.6–245.8)						
B2/B3 vs. B1	0.011 *	14.8 (1.85–118.7)						
Perianal disease	0.89	0.93 (0.33–2.62)						
Treatment with biologics	0.095	3.86 (0.79–18.91)			0.426	2.12 (0.34–13.38)		
CDAI	0.583	1.002 (0.99–1.01)						
HBI	0.418	1.048 (0.89–1.32)						
SES-CD	0.461	1.077 (0.88–1.31)						
ApaI			0.002 **	6.1 (1.9–19.59)	0.024 *	4.22 (1.21–14.7)	0.014 *	0.055 (0.006–0.56)
AA vs. Aa/aa	0.005 **	4.16 (1.55–11.2)						
AA vs. aa	0.01 **	17 (1.96–147.2)						
aa vs. AA/Aa	0.031 *	0.1 (0.12–0.81)						
BsmI								
BB vs. Bb/bb	0.263	1.95 (0.61–6.31)						
BB vs. bb	0.18	2.5 (0.66–9.46)						
bb vs. BB/Bb	0.256	0.56 (0.21–1.52)						
TaqI			0.55	1.63 (0.33–8.14)	0.649	1.48 (0.27–7.96)	0.085	3.57 (0.84–15.15)
TT vs. Tt/tt	0.038 *	3.25 (1.07–9.91)						
TT vs. tt	0.033 *	4.08 (1.2–14.89)						
tt vs. TT/Tt	0.147	0.47 (0.17–1.31)						
FokI								
FF vs. Ff/ff	0.79	1.15 (0.45–2.95)						
FF vs. ff	0.696	0.76 (0.19–3.04)						
ff vs. FF/Ff	0.524	1.52 (0.42–5.53)						

Model 1 includes all variables with *p*-value ≤ 0.05: age, behavior (B2/B3 vs. B1), ApaI (AA vs. Aa/aa), and TaqI (TT vs. Tt/tt). Model 2 includes all variables with a *p*-value ≤ 0.1: age, behavior (B2/B3 vs. B1), ApaI (AA vs. Aa/aa), TaqI (TT vs. Tt/tt), treatment with biologics, and active smoking. Model 3 includes all variables with a *p*-value ≤ 0.05, using the ApaI recessive genotype (aa) as a predictor: age, behavior (B2/B3 vs. B1), ApaI (aa vs. AA/Aa), and TaqI (TT vs. Tt/tt). CDAI: Crohn’s disease activity index; HBI: Harvey-Bradshaw index; SES-CD: simple endoscopic score for Crohn’s disease. * *p* < 0.05, ** *p* < 0.01.

## Data Availability

The data presented in this study are available on request from the corresponding author. The data are not publicly available as they involve human subjects, and their confidentiality and ethical considerations must be respected.

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
