# Peer review of "Genetic Polymorphisms (ApaI, FokI, BsmI, and TaqI) of the Vitamin D Receptor (VDR) Influence the Natural History and Phenotype of Crohn’s Disease"

_ijms, 2025, doi:10.3390/ijms26051848_

Round 1

Reviewer 1 Report

Comments and Suggestions for Authors

This study adds to the substantial literature on relationships between vitamin D related genes and Inflammatory bowel disease risk and phenotype. It is rather small for an analysis of this nature but will be of some interest and it is well written and discussed.

Some points need addressing:

1.     Abstract – need to reduce emphasis given to univariate analyses – eg start Results section with “On initial univariate analysis ….”

2.     Results/Discussion: The hospitalisation rate for UC is surprisingly high (51.4%) – no details are provided in “Methods” about how these data were captured – can the authors be sure that hospitalisation was because of disease-related symptoms rather than procedural needs such as colonoscopy or intravenous infusions? 

3.     Results/Discussion – the number of CD patients with perianal disease (n=11 in Table 1) (but see point 5 below) is very small and must be underpowered to look for any association – this point should be made firmly in discussion when the apparent negative result is discussed in contrast with ref 20.

4.     Results/Table 7 – I think “Behaviour B2/B2 vs B1” is meant to be “Behaviour B2/B3 vs B1”. 

5.     Results – perianal disease – Table 1 shows only 11 CD patients with perianal disease but Table 3 includes genotype data from 21 with perianal disease – is “11” in Table 1 a misprint?

6.     Discussion – the results of the univariate analyses need to be discussed with greater caution. At least 9 clinical endpoints (4 localisations, 3 behaviours, hospitalisation, surgery) have been analysed against 4 gene polymorphisms so a significant p value should have some Bonferroni or equivalent adjustment – eg perhaps a p value of <0.001 might be more convincingly significant if looking at these analyses – fine to use P<0.05 as cut off for inclusion into further multivariate analysis but not for discussion (without qualification) of the univariate analyses.

7.     Discussion – and Abstract – should therefore focus more firmly on the results of multivariate analysis that has shown a significant association between vitamin D receptor polymorphisms and risk for surgery in both CD and UC – although the multivariate analysis has also shown a link with hospitalisation risk this will of course overlap with surgery and is likely to be a less straightforward endpoint (see point 2 above). This is an interesting finding and the greater length of discussion given to the much less convincing associations found in the univariate analyses is likely to reduce the overall impact.

Author Response

Reviewer 1

This study adds to the substantial literature on relationships between vitamin D related genes and Inflammatory bowel disease risk and phenotype. It is rather small for an analysis of this nature but will be of some interest and it is well written and discussed.

Response to Reviewer 1: Thank you very much for your positive remarks and insightful comments.

Comment 1:  Abstract – need to reduce emphasis given to univariate analyses – eg start Results section with “On initial univariate analysis ….”

Response to comment 1: We have clarified in the abstract that bivariate analysis was used (page 1, line 17).

Comment 2: Results/Discussion: The hospitalisation rate for UC is surprisingly high (51.4%) – no details are provided in “Methods” about how these data were captured – can the authors be sure that hospitalisation was because of disease-related symptoms rather than procedural needs such as colonoscopy or intravenous infusions? 

Response to comment 2: We acknowledge that the hospitalization rate for UC patients in our study is higher than typically reported. In a systematic review and meta-analysis by Tsai et al., the 1-, 3-, and 5-year risks of UC-related hospitalization were 10.4%, 17.0%, and 21.5%, respectively, with considerable heterogeneity across cohorts (Tsai et al., 2022; doi:10.1007/s10620-021-07200-1). However, our study involved a long-term retrospective follow-up with a median duration of 10 years (IQR 4.5–18 years) for UC patients. Consistent with our findings, long-term studies such as that by Samuel et al. reported a cumulative probability of first hospitalization of 29.4% at 5 years, 38.7% at 10 years, and 49.2% at 20 years in UC patients (Samuel et al., 2013; doi:10.1097/MIB.0b013e31828c84c5). Given this extended follow-up period, it is not surprising that our hospitalization rate appears higher. The details of the retrospective follow-up duration for UC and CD patients in our cohort have been added to Table 1 (pages 2-3). We agree that it is important to consider the reasons for hospitalization. To clarify, our data specifically include only hospitalizations related to disease symptoms and flare-ups, excluding those associated with procedural needs such as colonoscopies or intravenous infusions. This distinction has been addressed in the Methods section (page 17, lines 483-487).

Comment 3: Results/Discussion – the number of CD patients with perianal disease (n=11 in Table 1) (but see point 5 below) is very small and must be underpowered to look for any association – this point should be made firmly in discussion when the apparent negative result is discussed in contrast with ref 20.

Response to comment 3: We appreciate your comment. The number of CD patients with perianal disease in our original manuscript was incorrectly stated as 11. In fact, 21 patients had perianal disease, and this error has been corrected (Table 1, page 3). We acknowledge that the small number of CD patients with perianal disease in our study likely resulted in insufficient statistical power to detect an association. To address this, we have explicitly emphasized this limitation in the discussion section when comparing our findings with those of Cusato et al (page 14, lines 345-347).

Comment 4: Results/Table 7 – I think “Behaviour B2/B2 vs B1” is meant to be “Behaviour B2/B3 vs B1”. 

Response to comment 4: Thank you for pointing that out. The label "Behavior B2/B2 vs B1" was an error, and it has been corrected to "Behavior B2/B3 vs B1" in Table 7 (page 12).

Comment 5: Results – perianal disease – Table 1 shows only 11 CD patients with perianal disease but Table 3 includes genotype data from 21 with perianal disease – is “11” in Table 1 a misprint?

Response to comment 5: Thank you for your observation. The "11" in Table 1 was indeed a misprint, and it has been corrected to reflect the accurate number of 21 CD patients with perianal disease

Comment 6: Discussion – the results of the univariate analyses need to be discussed with greater caution. At least 9 clinical endpoints (4 localisations, 3 behaviours, hospitalisation, surgery) have been analysed against 4 gene polymorphisms so a significant p value should have some Bonferroni or equivalent adjustment – eg perhaps a p value of <0.001 might be more convincingly significant if looking at these analyses – fine to use P<0.05 as cut off for inclusion into further multivariate analysis but not for discussion (without qualification) of the univariate analyses.

Response to comment 6: Our study is exploratory in nature, and now we have clarified this in the methods and discussion section to emphasize the need for further research on these associations (page 16, line 439). While the Bonferroni correction is considered a conservative approach in these studies, we have revised our bivariate analysis to strengthen the robustness of our results. For categorical variables, contingency tables were created for the three genotypes of each SNP and analyzed using Pearson's χ² test (3 × 2 and 3 × 3 tables). When significant results were obtained, further analysis was conducted using Fisher’s exact test (2 × 2 tables), comparing the genotype of interest against the combined group of the other two genotypes for the SNP. Bonferroni correction was then applied at this stage to adjust for multiple comparisons (page 18, lines 523-524). Additionally, Figures 1 and 2 have been revised to reflect these changes. Methods section has also been revised (page 18, lines 534-537).

Comment 7: Discussion – and Abstract – should therefore focus more firmly on the results of multivariate analysis that has shown a significant association between vitamin D receptor polymorphisms and risk for surgery in both CD and UC – although the multivariate analysis has also shown a link with hospitalisation risk this will of course overlap with surgery and is likely to be a less straightforward endpoint (see point 2 above). This is an interesting finding and the greater length of discussion given to the much less convincing associations found in the univariate analyses is likely to reduce the overall impact.

Response to comment 7: Abstract (page 1, lines 19-20-, 22-23) and discussion (pages 13-14, lines 309-323) sections have been altered accordingly.

Reviewer 2 Report

Comments and Suggestions for Authors

The topic is interesting. However, there are several concerns and as below.

Evidence should be added about the description of line 62-63.

The statistical differences of Table 3 should be clearly described. Which differences do each asterisk indicate?

In Figure 2, the categorization of genotypes seems to be inconsistent in Figure 1 and Figure 2. For example, why FF/Ff is accounted for same group despite the comparison of AA versus Aa/aa in Figure 1?

What is the rationale for the grouping that steroid-dependent and steroid-refractory are in same group? In the first place, the category of steroid-dependent is ambiguous.

There are no figures or tables corresponding to the description of line135-137.

For Tables 5 and 6, behavior and each SNPs have three comparison patterns. I assume that the multivariate analysis was conducted as a two-group comparison. Please specify which groups were compared in the analysis. In addition, the flows of the procedure of logistic regression analysis should be clearly described.

Normal allele showed the highest percentages of hospitalization among all groups in the analysis of the allele showed as Figure 3. These data appear to contradict the authors’ other data which suggest an association between SNPs and the onset or symptoms of CD and UC. Please discuss how this discrepancy can be interpreted in the Discussion section

Author Response

Reviewer 2

The topic is interesting. However, there are several concerns and as below.

Response to Reviewer 2: Thank you very much for your positive remarks and insightful comments.

Comment 1: Evidence should be added about the description of line 62-63.

Response to comment 1: Thank you for your observation. The reference has been added (page 2, line 66).

Comment 2: The statistical differences of Table 3 should be clearly described. Which differences do each asterisk indicate?

Response to comment 2: The asterisks indicate the following: *p<0.05 and **p<0.01. This explanation has been added to all relevant tables and figures for clarity.

Comment 3: In Figure 2, the categorization of genotypes seems to be inconsistent in Figure 1 and Figure 2. For example, why FF/Ff is accounted for same group despite the comparison of AA versus Aa/aa in Figure 1?

Response to comment 3: Thank you for your comment. As outlined in the methods section, for categorical variables (bivariate analysis), contingency tables were created for the three genotypes of each SNP and analyzed using the χ² test (3 × 2 and 3 × 3 tables). When significant results were obtained, further analysis was performed using Fisher’s exact test (2 × 2 tables) (enhanced precision for smaller samples), comparing the genotype of interest against the combined group of the other two genotypes for the SNP. The focus of the analysis was on the genotype that appears to influence disease patterns, which could correspond to either the dominant or the recessive genotype (please also refer to our response to your last comment). Please note that in the revised manuscript, we applied Bonferroni correction to adjust for multiple comparisons (page 18, lines 523-524).

Comment 4: What is the rationale for the grouping that steroid-dependent and steroid-refractory are in same group? In the first place, the category of steroid-dependent is ambiguous.

Response to comment 4 The rationale for grouping steroid-dependent and steroid-refractory patients together lies in their shared characteristic of inadequate response to steroids. While it can be challenging to clearly distinguish between the two groups, as the response to steroids can vary over time or in different clinical settings, both conditions involve steroid resistance or dependence. Additionally, the overlap in treatment strategies complicates their categorization in some instances. However, by strictly following the ECCO definitions, we aimed to improve consistency.

Comment 5: There are no figures or tables corresponding to the description of line 135-137.

Response to comment 5: This line has been deleted (page 6, lines 139-142..

Comment 6: For Tables 5 and 6, behavior and each SNPs have three comparison patterns. I assume that the multivariate analysis was conducted as a two-group comparison. Please specify which groups were compared in the analysis. In addition, the flows of the procedure of logistic regression analysis should be clearly described.

Response to comment 6: Indeed, the multivariate analyses (Tables 5, 6, and 7) use two-group comparisons for disease behavior and each SNP. Below each table, we have already specified the comparison groups for each variable in the multivariate models, along with the included variables. For example, in Table 7, Model 1 compares behavior (B2/B3 vs. B1), ApaI (AA vs. Aa/aa), and TaqI (TT vs. Tt/tt). A backward stepwise approach was applied, incorporating all variables with p ≤ 0.05 from the univariate analysis. To validate the results, additional multivariate models were conducted using an alternative p-value threshold (0.1) for variable inclusion. We have now further clarified this procedure in the Methods section (page 18, lines 534-537).

Comment 7: Normal allele showed the highest percentages of hospitalization among all groups in the analysis of the allele showed as Figure 3. These data appear to contradict the authors’ other data which suggest an association between SNPs and the onset or symptoms of CD and UC. Please discuss how this discrepancy can be interpreted in the Discussion section.

Response to comment 7: Thank you for your comment. The designation of a dominant or recessive trait for each SNP does not inherently imply that the dominant allele is protective or the recessive allele detrimental. These classifications are often a matter of definition and can sometimes be arbitrary. Additionally, as shown in Table 1, the bb/tt genotypes are more common in our cohort than the BB/TT genotypes, despite the latter being defined as dominant.

What is important is that the allelic variations are in Hardy-Weinberg equilibrium, ensuring that the observed genotype distributions are not biased by population stratification or genotyping errors. We have added a relevant comment in the results section for clarity (pages 3-4, lines 92-94). To ensure transparency, Table 2 provides the definition of each VDR SNP along with the corresponding allelic variations.

In our bivariate analysis, the A allele is associated with penetrating disease and inadequate steroid response, suggesting a more severe disease course. This is further supported by our multivariate analysis, which consistently identifies the AA genotype as an independent predictor of IBD-related surgery and hospitalization. We have now revised the discussion to clarify this further (pages 15-16, lines 421-432).

Round 2

Reviewer 2 Report

Comments and Suggestions for Authors

The manuscript has been revised.